# Variation in (Hyper)Polarizability of H_2_ Molecule in Bond Dissociation Processes Under Spatial Confinement

**DOI:** 10.3390/molecules30010009

**Published:** 2024-12-24

**Authors:** Paweł Lipkowski, Wojciech Bartkowiak

**Affiliations:** Department of Physical and Quantum Chemistry, Wrocław University of Science and Technology, 50-370 Wrocław, Poland; pawel.lipkowski@pwr.edu.pl

**Keywords:** spatial confinement, (hyper)polarizability, hydrogen molecule, bond dissociation

## Abstract

We report the results of calculations of the linear polarizability and second hyperpolarizability of the H_2_ molecule in the bond dissociation process. These calculations were performed for isolated molecules, as well as molecules under spatial confinement. The spatial confinement was modeled using the external two-dimensional (cylindrical) harmonic oscillator potential. In contrast to the recently investigated polar LiH molecule, it was shown that the spatial confinement significantly diminishes the linear and nonlinear response of H_2_ for each interatomic (H-H) distance.

## 1. Introduction

At the outset, it should be noted that the aim of the present work is to supplement our previous two papers [1,2]. In these studies, the most important findings highlighted how spatial confinement (modeled using the two-dimensional (cylindrical) harmonic oscillator potential) may significantly enhance the linear polarizability (*α*) and nonlinear electrical response defined by the first hyperpolarizability (*β*), second hyperpolarizability (*γ*), and two-photon absorption cross-section (*δ*) of molecules during the dissociation process. The obtained results were restricted to the model LiH molecule. The changes in *α*, *β*, *γ*, and *δ* in the bond dissociation process of LiH were investigated by the finite-field (FF) approach using the highly accurate multiconfiguration self-consistent field (MCSCF) and response function formalism [1,2]. In particular, it was shown that the nonlinear optical response strength (*β*, *γ,* and *δ*) may increase by orders of magnitude.

Due to the specific electronic structure of LiH, for which the first excited state has outstanding charge-transfer (CT) character, the explanation of the behavior of the hyperpolarizabilities (especially *β*) and *δ* as a function of the bond length, as well as the strong enhancement effect on nonlinear responses (both for the isolated molecule and after including spatial confinement), was possible on the basis of the simple two-level models. On the other hand, the multilevel analysis of this phenomenon (for isolated molecular systems) was provided by Nakano and Champagne, who employed the valence configuration interaction model of asymmetric open-shell singlet molecules combined with few-state models [3]. The variations in the dipole moment (*µ*) and polarizability (*α*) during the bond dissociation process of LiH, including the effects of spatial confinement, were investigated by Klobukowski et al. [4].

Inspired by the results obtained for LiH, in this paper, the longitudinal *α* and *γ* (components along the axis of molecules) of the H_2_ molecule as a function of intermolecular distance (R) are calculated taking into account the spatial confinement. The main goal of this paper is to show the potential impact of the spatial confinement on linear (*α*) and nonlinear (*γ*) electrical response during the bond dissociation process of H_2_, representing a family of single-bond (centrosymmetric) molecules on the background of the strong polar LiH molecule.

Because of its simple structure and small size, the H_2_ molecule has been extensively studied in the context of how spatial confinement affects molecular structure and valence bonds [5,6,7,8,9,10,11,12,13,14,15,16,17,18,19,20]. These studies have been conducted using various model confining potentials differing in mathematical structure, such as penetrable, impenetrable (infinite-walled), spherical, cylindrical, spheroidal, icosahedral, and so on. On the other hand, there is a limited number of studies that quantitatively consider the problem of the confinement effect on the dissociation process of molecular hydrogen [6,9,10,13,16,19]. Of particular importance is the work by LeSar and Herschbach, which presents the calculations of the *α* tensor components of the H_2_ molecule enclosed within the nonpenetrable spheroidal box [6]. These authors have analyzed the variation in *α* as a function of the box size, as well as internuclear distance. The general conclusions that emerge from these studies are as follows: (a) there is a notable reduction in the H-H bond length and the bond becoming stiffer as the confinement strength increases, (b) vibrational frequency increases relative to the free molecule, (c) with increasing confinement strength, the absorption and emission of vibronic bands also become blue shifted, (d) the total energy of the confined molecule increases under the confinement, with the most significant effect seen in kinetic energy, and (e) in the presence of spatial confinement, the *α* and *γ* values of H_2_ are significantly smaller relative to that of a free H_2_ molecule. Note that most of the observations cited here also apply to other molecular systems [21].

## 2. Theory

The choice of method for calculations was based on the recently obtained results by Miliordos and Hunt [22]. These authors applied, after detailed analysis, the configuration interaction singles and doubles (CISD) method with the extended d-aug-cc-pV6Z basis set for various H-H separations (R). It has been shown that the static *α* and *γ* values at this level of theory agree well with high-accuracy reference results (see [22] and the references therein). In order to assess the effect of the basis set on the energy and different electric properties, including *α* and *γ*, these authors have used various correlation-consistent basis sets [22]. It has been shown that the d-aug-cc-pV6Z and t-aug-cc-pV6Z values for the properties are converged within 1%. On the other hand, in contrast to the t-aug-cc-pV6Z basis, in the case of the d-aug-cc-pV6Z basis, linear dependence at shorter H-H distances is not observed [22]. It should be noted that achieving convergence with respect to basis functions in the case of molecular (hyper)polarizabilities is much more difficult than in the case of energy. Hence, the results presented in our paper were obtained at the same level of theory, with one exception, i.e., additional mid-bond functions were not included. However, this omission has a marginal impact on the quality of results. It should be noted that our values of *α* and *γ* are in good agreement with those obtained by Miliordos and Hunt [22] for the range of H-H distances taken into account in our investigations.

The dominant longitudinal components of the static *α (α_xx_*) and *γ* (*γ_xxxx_*) values were evaluated using the finite-field (FF) approach by calculating the total energies under the finite electric fields and applying numerical differentiation. The FF technique is based on the Taylor expansion of the energy (*E*) of a molecular system in the presence of an external and uniform electric field (**F**):*E*(**F**) = *E*_0_ − *µ_α_*F*_α_* − (1/2) *α_αβ_* F*_α_*F_β_ − (1/6)*β_αβγ_*F*_α_*F*_β_*F*_γ_* − (1/24)*γ_αβγδ_*F*_α_*F*_β_*F*_γ_*F*_δ_* + … (1)

Through this expression, the components of linear polarizability, as well as first- and second-order hyperpolarizability tensors, are defined: *α_αβ_*, *β_αβγ_*, and *γ_αβγδ_*. The components of **F** are included in the molecular Hamiltonian and, based on the above expansion, the molecular (hyper)polarizabilities are defined by subsequent derivatives of the energy with respect to the electric field, i.e., second-order, third-order, and fourth-order derivatives correspond to *α*, *β*, and *γ*, respectively. During the computations, the self-consistent field (SCF) convergence threshold and the CISD energy convergence criterion were set to 10^−12^ and 10^−10^ a.u., respectively.

The numerical differentiation was performed based on the Romberg–Rutishauser algorithm [23], applying the electric field amplitudes ±2^n^ h, where h = 0.0002 a.u. and n = 0, 1, … 6, thereby minimizing the contamination of higher power (in the field) terms. All computations were carried out with the aid of Gaussian 16, Revision C.01, package [24]. The results of our calculations are presented in atomic units (a.u.).

In this work, the spatial confinement was modeled by the two-dimensional harmonic oscillator (HO) potential of cylindrical symmetry:(2)$${\hat{V}}_{c}\left({\vec{r}}_{i}\right)\;=\;\frac{1}{2}\omega^{2}{{\vec{r}}_{i}}^{2}=\frac{1}{2}\omega^{2}\left(x_{i}^{2}\;+\;y_{i}^{2}\right)$$
and was applied to mimic the effect of orbital compression. In Equation (2), the *ω* parameter, which is the frequency of the oscillator with the mass equal to *m_e_*, defines the strength of spatial confinement. The HO potential is included in the clamped nuclei Hamiltonian of an isolated *n*-electron (in our case, 2-electron) system (${\hat{H}}^{0}$): $\hat{H}\;=\;{\hat{H}}^{0}\;+\;{\hat{V}}_{c}(\vec{r})$. In all computations, the principal axis of the HO potential overlaps with the molecular axis of H_2_, which is assumed to be the *x*-axis.

In summary, the two-dimensional HO potential has the following features: (a) The HO potential acts only on electrons. The effect of confinement on nuclei is accounted for indirectly through its effect on electrons (it is well known that this approximation is formally correct in both quantitative and qualitative terms), and (b) the HO potential belongs to the group of repulsive potentials. This type of repulsive potential accounts mainly for the effects arising from the Pauli exclusion principle, though it may overestimate the effect of compression or high pressure. Thus, this type of confinement is often referred to as “pure” confinement; (c) because of its cylindrical symmetry, the HO potential serves as a simplified representation of the nanotube-like confining cages, and (d) the influence of the HO potential on the molecular structure is in many aspects similar to that a parallel magnetic field. Because the simple potentials allow a straightforward description of the real confinement environments, the application of such potentials is a very attractive concept. A more comprehensive presentation of the formal and computational aspects associated with the HO potential and other model potentials can be found in references [10,21,25,26,27,28,29,30].

## 3. Results and Discussion

Our results for the isolated H_2_ molecule only confirm findings from other previous works considering the behavior of *α* and *γ* (see [22] and the references cited therein). The curves of potential energy, as well as numerical values of the total energy, *α*, and *γ*, are presented in the Appendix A. The dependence of *α* and *γ* on internuclear distance R (ranging from 1.04 a.u. to 6.61 a.u.) is nonmonotonic (see Figure 1 and Figure 2).

As the H-H bond is elongated, the values of *α* and *γ* increase, and then these quantities attain maximum values (*α_max_
*= 18.29 a.u. and *γ_max_
*= 15,761 a.u.). Further stretching of the H-H bond leads to a significant decrease in *α* and *γ*. It should be noticed that the maximum value of *γ* is obtained for a relatively larger value of internuclear distance (R = 4.16 a.u.) in comparison with *α* (R = 3.40 a.u.). For shorter H-H separations (R < R_e_), the values of *α* and *γ* decrease significantly.

Additionally, a substantial enhancement of the nonlinear electrical response (*γ*) is observed. The maximum *γ* value is approximately two orders of magnitude larger than that at the equilibrium distance (R_e_ = 1.40 a.u.). At the equilibrium distance, *α* and *γ* are equal to 6.40 a.u. and 689 a.u., respectively. In the dissociation limit, both *α* and *γ* reach values equal to 2*α_H_* and 2*γ_H_*, where *α_H_* and *γ_H_* mean values for the isolated hydrogen atom. As was shown by Nakano et al., the behavior of the longitudinal *α* and *γ* of the H_2_ molecule as a function of the intermolecular distance (R) is directly connected with its singlet diradical character [31]. Note that the diradical character is a measure of the chemical bond [32]. In particular, the explanation of the significant enhancement values of the longitudinal *α* and *γ* at intermediate diradical character, on this basis, is strongly proved. It is not necessary to quote these arguments and theories again. The interested reader is kindly referred to the rich literature on this field [31,32,33,34]. We can also interpret the observed maxima in Figure 1 and Figure 2 as a result of the expansion of electronic charges (density) that takes place when hydrogen atoms get closer to each other from a long distance, which is followed by a contraction at smaller values of R [35].

In order to investigate the influence of spatial confinement on *α* and *γ*, we consider *ω* values of the OH potential in the range 0–0.8 a.u. The value of *ω* equal to 0.8 means an extremely large strength of spatial confinement (see discussion in ref. [28]). First, it should be noticed that the reduction in the equilibrium interatomic distances (R_e_) is observed in the presence of spatial confinement (R_e_(*ω* = 0.0) = 1.40 a.u., R_e_(*ω* = 0.10) = 1.39 a.u., R_e_(*ω* = 0.40) = 1.27 a.u., and R_e_(*ω* = 0.80) = 1.14 a.u.). As was pointed out in the introduction, this shortening of the covalent bond induced by spatial confinement (described by repulsive external potentials just like OH) is prevalent in molecular systems. It is connected with the changes in the distribution of electron density which, under the influence of confining potential, accumulates in the spaces between atoms, and finally, the repulsive force between nuclei becomes smaller.

As shown in Figure 1 and Figure 2, there is a significant decrease in the *α* and *γ* values upon increasing the confinement strength. This observation refers to all R values. In a qualitative sense, these results are well supported by the findings obtained by LeSar and Herschbach for *α* of H_2_ [6]. Moreover, in contrast to the unrestricted (*ω* = 0) H_2_ molecule, the maxima are shifted towards smaller values of R, the greater the strength of the spatial confinement. The maximum values of *α* (*α_max_*(*ω* = 0.10) = 17.03 a.u., *α_max_*(*ω* = 0.40) = 11.15 a.u., *α_max_*(*ω* = 0.80) = 7.33 a.u.) are reached for the following distances (R): 3.21 a.u., 2.83 a.u., and 2.65 a.u., respectively. In the case of *γ*, the maximum values (*γ_max_*(*ω* = 0.10) = 12,475 a.u., *γ_max_*(*ω* = 0.40) = 3835 a.u., and *γ_max_*(*ω* = 0.80) = 1275 a.u.) are observed for longer distances (R): 4.15 a.u., 3.59 a.u., and 3.40 a.u., respectively. In contrast to the LiH molecule investigated in the same context in our earlier works [1,2], the obtained results for H_2_ fit into the general knowledge in this field. It is well established that the presence of orbital compression (deformation) caused by the repulsive external potentials leads to a decrease in the values of *α* and *γ* for molecular systems and atoms [21]. This was not obvious prior to our research. We should remember that the above observations refer usually to molecules located at the minimum of the potential energy curve, in other words, for the equilibrium interatomic distances (R_e_). Here, we also have clearly demonstrated this effect for *γ* during the dissociation process of the H_2_ molecule. It should be remembered that *α* is proportional to the volume of an atom or molecule. The effective volume (space for electron density) is notably reduced when we enable space restriction. On the other hand, the confinement (understood as in this work) would cause an increase in the gap between frontier orbitals (HOMO and LUMO) relative to that of an unconfined atom or molecule. On the basis of the simple two-level model, derived within the sum-over-orbitals (SOO) method, it can be shown that *α* is inversely proportional to the difference between the energy of the LUMO and HOMO orbitals [36]. Hence, we have two perspectives on explaining the behavior of *α* (a similar situation is in the case of *γ*) upon confinement modeled by the OH potential.

Summing up, the most important question of this work was to explore the bond-length dependence of linear and nonlinear electrical response of spatially restricted H_2_ molecules. The calculations for *γ*, in this context, have never been considered in the literature before. The results of our investigations clearly indicate a significant reduction in the values of *α* and *γ* of molecular hydrogen, not only at its equilibrium bond length (R_e_) but also for R > R_e_ and R < R_e_ upon confinement. In contrast to the H_2_ molecule, our previous calculations for the model LiH molecule (based on the same OH potential) showed a very different picture [1]. Firstly, when the bond length of the LiH molecule embedded in the OH potential is strongly stretched, the *α*, *γ*, and also *β* (because of symmetry, *β* = 0 for H_2_) can increase by several orders of magnitude. Secondly, the course of dependencies of (hyper)polarizabilities as a function of R is also definitively different. In the case of LiH, the maxima and minima of *α*, *β*, and *γ* as a function of Li-H separations are observed for R > R_e_ (in the intermediate dissociation region). At this stage, only two model molecules with single covalent bonds and distinct electronic structures have been examined. Therefore, it is premature to make broad generalizations about the behavior of other molecules during dissociation. However, we believe that, given the findings of this study and prior work [1,2], a meaningful step toward understanding these effects has been achieved.

## Figures and Tables

**Figure 1 molecules-30-00009-f001:**
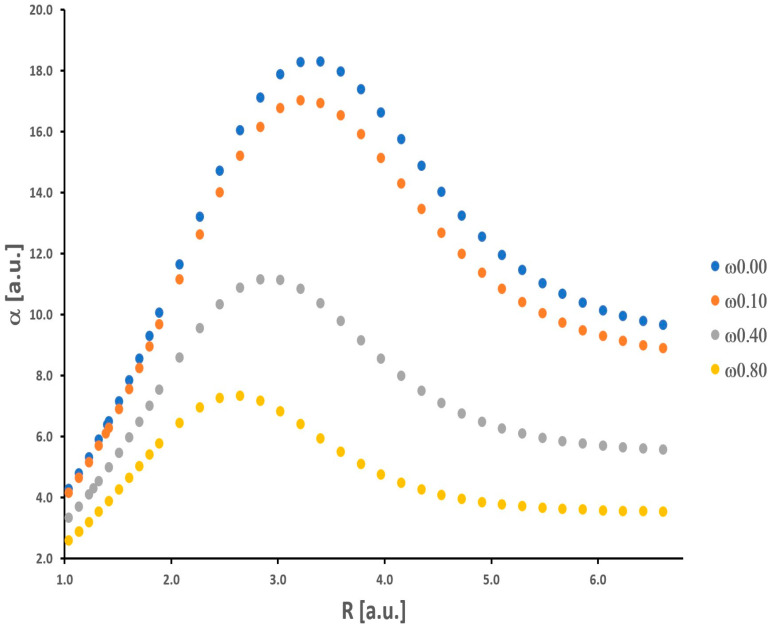
Polarizability of the H_2_ molecule obtained as a function of the internuclear distance.

**Figure 2 molecules-30-00009-f002:**
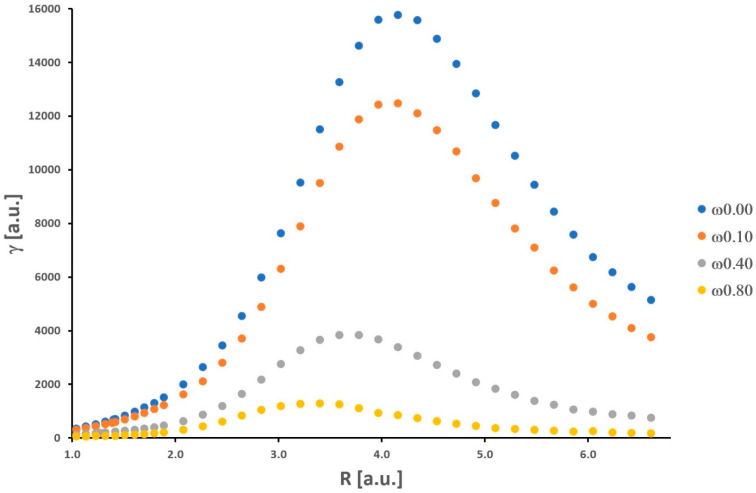
Second hyperpolarizability of the H_2_ molecule obtained as a function of the internuclear distance.

## Data Availability

Data are contained within the article and Appendix A.

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
