# Peer review of "Variation in (Hyper)Polarizability of H2 Molecule in Bond Dissociation Processes Under Spatial Confinement"

_molecules, 2024, doi:10.3390/molecules30010009_

Round 1
Reviewer 1 Report
Comments and Suggestions for Authors
The manuscript “Variation of (hyper)Polarizability of the H2 molecule in the Bond Dissociation Processes under Spatial Confinement” by Lipkowski and Bartowiak reports a computational study aimed at describing the effet of bond elongation on the linear polarizability (alpha) and second hyperpolarizability (gamma) of both the unperturbed and spatially confined H2 molecule. The values of alpha and gamma were found to be significantly affected (reduced) by the presence of external confinement at – essentially – whatever interatomic distance. The manuscript is relatively well written, very well inserted into the scientific context and the results are consistent and obtained with an (apparently) solid computational scheme.For these reasons I suggest the publication of this study without substantial changes. I only have two methodological (formal) remarks. The descritpion of the theoretical strategy, in my opinioxn, can be a little bit expanded in the main text without leaving the reader with the need to go through the literature. Probably my question may appear trivial but I have to ‘put myself in the shoes of the non-expert reader’. In particular (i) it is not clear to me whether the CISD calculations were used to provide a basis-set used, subsequently, for obtaining the actual eigenstates of the Hamiltonian in the presence of HO potential or if the CISD is construced using the ground state eigenstate in the presence of the perturbation; (ii) it is also not very clear whether the authors, considering the different interatomic distances and the relatively high fields, have limited their attention on the singlet H2 (maghetic) state.
Author Response
The manuscript “Variation of (hyper)Polarizability of the H2 molecule in the Bond Dissociation Processes under Spatial Confinement” by Lipkowski and Bartowiak reports a computational study aimed at describing the effet of bond elongation on the linear polarizability (alpha) and second hyperpolarizability (gamma) of both the unperturbed and spatially confined H2 molecule. The values of alpha and gamma were found to be significantly affected (reduced) by the presence of external confinement at – essentially – whatever interatomic distance. The manuscript is relatively well written, very well inserted into the scientific context and the results are consistent and obtained with an (apparently) solid computational scheme.For these reasons I suggest the publication of this study without substantial changes. I only have two methodological (formal) remarks. The descritpion of the theoretical strategy, in my opinioxn, can be a little bit expanded in the main text without leaving the reader with the need to go through the literature. Probably my question may appear trivial but I have to ‘put myself in the shoes of the non-expert reader’. In particular (i) it is not clear to me whether the CISD calculations were used to provide a basis-set used, subsequently, for obtaining the actual eigenstates of the Hamiltonian in the presence of HO potential or if the CISD is construced using the ground state eigenstate in the presence of the perturbation; (ii) it is also not very clear whether the authors, considering the different interatomic distances and the relatively high fields, have limited their attention on the singlet H2 (maghetic) state.
Comments 1: In particular (i) it is not clear to me whether the CISD calculations were used to provide a basis-set used, subsequently, for obtaining the actual eigenstates of the Hamiltonian in the presence of HO potential or if the CISD is construced using the ground state eigenstate in the presence of the perturbation;
Response 1: Thank you for your attention. As indicated in our article, the choice of method and basis set for calculations was based on the recently obtained results by Miliordos and Hunt [22]. These authors applied, after detailed analyzes, the configuration interaction singles and doubles (CISD) method with the extended d-aug-cc-pV6Z basis set for various H–H separations (R). Hence, the answer is affirmative. However, the CISD calculations were used for to assess the quality of the basis set for the isolated hydrogen molecule. In our opinion, the selected basis set is saturated enough also for calculations of the electric properties in the presence of the HO potential. We hope this is clear from our publication. (We have slightly expanded the description of the methodology in Section 2)
Comments 2: (ii) it is also not very clear whether the authors, considering the different interatomic distances and the relatively high fields, have limited their attention on the singlet H2 (maghetic) state.
Response 2: We do not know whether we have understood this reviewer's comment correctly. The choice of electric field amplitudes in the FF calculations is a key issue determining the correctness of the obtained results. In this case, in our opinion, the best solution is to use the Romberg-Rutishauser algorithm [23]. Only in this context were we interested in the applied electric fields. In fact, we considered the ground singlet state only in terms of electrical properties. the problem raised by the reviewer is interesting but requires longer analyzes that we will conduct in the future.
Reviewer 2 Report
Comments and Suggestions for Authors
In this work the authors have performed very accurate electronic structure calculations of hyperpolarizability of the confined H2 molecule using a sophisticated computational method. The manuscript may be publishable once the following points would be addressed by the authors.
1. I would like to know the calculated potential energy curves as a function of R although the authors have given some explanations on page 5. Such information should be more useful for readers. In addition to this, I recommend the authors add the variation of the “diradical factor” as a function of R.
2. Please describe the detailed explanation for the reason why the d-aug-cc-pV6Z basis set was chosen in this work. The description “after detailed analyzed” on Page 2 may be insufficient. I think that the converged electronic energy can be obtained with relatively smaller basis sets but I am not sure about (hyper)polarizabilities as well as other electronic properties.
3. If possible, I recommend the authors to add some descriptions on the relation between experimental measurements (or related phenomenon in nature) and the present electronic structure calculations.
Author Response
In this work the authors have performed very accurate electronic structure calculations of hyperpolarizability of the confined H2 molecule using a sophisticated computational method. The manuscript may be publishable once the following points would be addressed by the authors.
Comments 1: I would like to know the calculated potential energy curves as a function of R although the authors have given some explanations on page 5. Such information should be more useful for readers. In addition to this, I recommend the authors add the variation of the “diradical factor” as a function of R.
Response 1: Thank you for pointing this out. As suggested by the reviewer, we have included a graph of potential energy curves in Supplementary Materials (Fig. S1) [citation added in line 130 in the manuscript]. The diradical factor (DF) as a function of R for the H2 molecule is shown in refs. [31,33], which are cited in our manuscript. Hence, we decided not to show this relationship again. However, in the future, the behavior of DF as a function of confinement strength will be the subject of our research. This is a very valuable suggestion!
Comments 2: Please describe the detailed explanation for the reason why the d-aug-cc-pV6Z basis set was chosen in this work. The description “after detailed analyzed” on Page 2 may be insufficient. I think that the converged electronic energy can be obtained with relatively smaller basis sets but I am not sure about (hyper)polarizabilities as well as other electronic properties.
Response 2: As indicated in our article, the choice of method and basis set for calculations was based on the recently obtained results by Miliordos and Hunt [22]. In order to assess the effect of the basis set on the energy and different electric properties including α and γ, these authors have used various correlation-consistent basis sets. It has been shown that the d-aug-cc-pV6Z and t-aug-cc-pV6Z values for the properties are converged within 1%. On the other hand, in contrast to the t-aug-cc-pV6Z basis, in the case of the d-aug-cc-pV6Z basis linear dependence at shorter H-H distances are not observed. It should be noted that achieving convergence with respect to basis functions in the case of molecular (hyper)polarizabilities is much more difficult than in the case of energy.
We have added the following sentence in the Section 2 (lines 71-78):
“In order to assess the effect of the basis set on the energy and different electric properties including α and γ, these authors have used various correlation-consistent basis sets. It has been shown that the d-aug-cc-pV6Z and t-aug-cc-pV6Z values for the properties are converged within 1%. On the other hand, in contrast to the t-aug-cc-pV6Z basis, in the case of the d-aug-cc-pV6Z basis linear dependence at shorter H-H distances are not observed. It should be noted that achieving convergence with respect to basis functions in the case of molecular (hyper)polarizabilities is much more difficult than in the case of energy.”
Comments 3: If possible, I recommend the authors to add some descriptions on the relation between experimental measurements (or related phenomenon in nature) and the present electronic structure calculations.
Response 3: The issue raised by the reviewer in this point is very interesting. After deep analysis, we must conclude that we are unable to indicate a direct connection between our calculations and the experiments that can be performed. However, progress in mechanochemistry and the development of instrumental methods related to the manipulation of molecules give hope that such subtle experiments will be possible in the future.